# Pathophysiology of Group 3 Pulmonary Hypertension Associated with Lung Diseases and/or Hypoxia

**DOI:** 10.3390/ijms26020835

**Published:** 2025-01-20

**Authors:** Kazufumi Nakamura, Satoshi Akagi, Kentaro Ejiri, Satoshi Taya, Yukihiro Saito, Kazuhiro Kuroda, Yoichi Takaya, Norihisa Toh, Rie Nakayama, Yuki Katanosaka, Shinsuke Yuasa

**Affiliations:** 1Department of Cardiovascular Medicine, Okayama University Graduate School of Medicine, Dentistry and Pharmaceutical Sciences, Okayama 700-8558, Japan; akagi-s@cc.okayama-u.ac.jp (S.A.); kejiri1@s.okayama-u.ac.jp (K.E.); me19049@s.okayama-u.ac.jp (S.T.); p5438a3l@okayama-u.ac.jp (Y.S.); pzu89o22@okauama-u.ac.jp (K.K.); takayayoichi@okayama-u.ac.jp (Y.T.); norihisa.toh@okayama-u.ac.jp (N.T.); rnakayama@okayama-u.ac.jp (R.N.); yuasa@okayama-u.ac.jp (S.Y.); 2Center for Advanced Heart Failure, Okayama University Hospital, Okayama 700-8558, Japan; 3Department of Pharmacy, Kinjo Gakuin University, Nagoya 463-8521, Japan; katanosaka@kinjo-u.ac.jp; 4Graduate School of Pharmaceutical Sciences, Kinjo Gakuin University, Nagoya 463-8521, Japan

**Keywords:** group 3 pulmonary hypertension, hypoxic pulmonary vasoconstriction, pulmonary vascular remodeling

## Abstract

Pulmonary hypertension associated with lung diseases and/or hypoxia is classified as group 3 in the clinical classification of pulmonary hypertension. The efficacy of existing selective pulmonary vasodilators for group 3 pulmonary hypertension is still unknown, and it is currently associated with a poor prognosis. The mechanisms by which pulmonary hypertension occurs include hypoxic pulmonary vasoconstriction, pulmonary vascular remodeling, a decrease in pulmonary vascular beds, endothelial dysfunction, endothelial-to-mesenchymal transition, mitochondrial dysfunction, oxidative stress, hypoxia-inducible factors (HIFs), inflammation, microRNA, and genetic predisposition. Among these, hypoxic pulmonary vasoconstriction and subsequent pulmonary vascular remodeling are characteristic factors involving the pulmonary vasculature and are the focus of this review. Several factors have been reported to mediate vascular remodeling induced by hypoxic pulmonary vasoconstriction, such as HIF-1α and mechanosensors, including TRP channels. New therapies that target novel molecules, such as mechanoreceptors, to inhibit vascular remodeling are awaited.

## 1. Introduction

Pulmonary hypertension (PH) is a global health problem. Current estimates put the prevalence of PH at least 1% of the world’s population, which increases up to 10% in individuals aged more than 65 years [1,2]. Globally, left heart disease is the leading cause of PH [2]. Lung disease is the second most common cause [2]. Mild PH is common in advanced parenchymal and interstitial lung disease. It has been reported that ~1–5% of patients with advanced chronic obstructive pulmonary disease (COPD) with chronic respiratory failure or candidates for lung volume reduction surgery or lung transplantation (LTx) have a mean pulmonary artery pressure (mPAP) of >35–40 mmHg [3,4,5]. In idiopathic pulmonary fibrosis, 8–15% of patients have an mPAP of ≥25 mmHg at the initial examination, with a higher prevalence reported in the advanced (30–50%) and end stages (>60%) [3,5].

### 1.1. Definition of Pulmonary Hypertension

Following the first World Symposium on Pulmonary Hypertension (WSPH), held in Geneva in 1973, PH was defined as an mPAP of ≥25 mmHg, as measured by means of right heart catheterization in the supine position at rest [6,7]. This definition was also adopted at the WSPH held in Nice in 2013 [8].

In 1961, a report by the World Health Organization Expert Committee on Chronic Cor Pulmonale stated that mPAP does not usually exceed 15 mmHg at rest and never exceeds 20 mmHg [9]. Therefore, at the sixth WSPH, held in Nice in 2018, it was proposed that the definition of PH be changed to a mean PAP of >20 mmHg [7]. In 2022, the European Society of Cardiology (ESC)/European Respiratory Society (ERS) guidelines for the diagnosis and treatment of PH specified that PH is defined by an mPAP of >20 mmHg at rest [3]. This is supported by the results of studies in which the upper limit of normal pulmonary arterial pressure (PAP) was evaluated in healthy subjects and studies in which the prognostic relevance of elevated PAP was investigated [10,11].

COPD is often associated with PH. In a study of 120 patients with severe COPD undergoing right heart catheterization, 85.8% of the patients had an mPAP of ≥20 mmHg [12]. On the other hand, only 5.0% of patients with severe PH had an mPAP of ≥35 mmHg. In a retrospective analysis of right heart catheterization data from 4930 COPD cases, 47.6% of the cases had an mPAP of ≥25 mmHg [13]. On the other hand, only 4% of the cases had severe PH with an mPAP of ≥35 mmHg. Complications of PH serve as poor prognostic factors. In a study in which 53 COPD patients underwent right heart catheterization, mPAP increased in proportion to the severity of COPD, and patients with an mPAP of ≥20 mmHg had a poorer prognosis than those without PH [14]. These reports indicate that COPD is associated with a certain percentage of PH, but the severity of PH is low, and the prognosis for patients with an mPAP of ≥20 mmHg is poor. Thus, it is reasonable to also define PH associated with lung disease as an mPAP of ≥20 mmHg.

### 1.2. Clinical Classification of Pulmonary Hypertension

PH is classified into the following five groups: PAH (group 1), PH associated with left heart disease (group 2), PH associated with lung diseases and/or hypoxia (group 3), PH associated with pulmonary artery obstructions (group 4), and PH with unclear and/or multifactorial mechanisms (group 5) [3,7].

As mentioned above, mild PH is common in patients with advanced COPD, and patients with PH have a poorer prognosis than those without PH [12,13,14]. PH is often observed in patients with idiopathic pulmonary fibrosis (IPF) and has also been shown to be a poor prognostic factor for interstitial lung disease (ILD) [15]. An mPAP of ≥25 mmHg has been reported in 30–50% of patients with advanced IPF [5].

PH associated with lung diseases—such as COPD and interstitial pneumonia—and/or hypoxia is classified as group 3 in the clinical classification of PH [3,7], and PH is known to be a poor prognostic factor for various respiratory diseases [15,16,17]. Several selective pulmonary vasodilators, including prostacyclin (PGI_2_), phosphodiesterase type-5 (PDE5) inhibitors, soluble guanylate cyclase stimulators, and endothelin receptor antagonists (ERAs), have been developed for the treatment of group 1 pulmonary arterial hypertension (PAH), the prognosis of which has improved dramatically [18,19]. On the other hand, the efficacy of selective pulmonary vasodilators for group 3 PH is still unclear, and it is associated with a poor prognosis [5,20]. Understanding the pathogenesis of group 3 PH is important for developing new treatment strategies. In this article, we discuss the various mechanisms underlying the pathogenesis of group 3 PH.

## 2. Pathogenesis of Group 3 Pulmonary Hypertension

There are several mechanisms that cause PH associated with lung diseases and/or hypoxia (Table 1).

This review discusses the mechanisms of group 3 PH listed in Table 1, particularly pulmonary vasoconstriction and subsequent vascular remodeling, with a focus on mechanosensors.

## 3. Causal Factors of Pathogenesis of Group 3 PH

### 3.1. Hypoxic Pulmonary Vasoconstriction

In the systemic circulation, during hypoxic exposure, blood vessels to the respective organs dilate to protect the tissues from hypoxia. In the pulmonary circulation, on the other hand, when the partial pressure of alveolar oxygen is decreased, the vascular smooth muscle of microvessels adjacent to the alveoli contracts. This is a physiological response called hypoxic pulmonary vasoconstriction to reduce the exacerbation of hypoxemia [21]. The reduction in blood flow to the alveoli with low alveolar air oxygen partial pressure increases blood flow to the alveoli with higher alveolar air oxygen partial pressure, correcting the ventilation–perfusion ratio (V/Q) mismatch. When this occurs persistently or over a large vessel area, pulmonary vascular resistance (PVR) can increase, leading to PH [22].

Acute vasoconstriction is biphasic, consisting of an initial transient contraction (phase I) and a subsequent sustained contraction (phase II), which involves increased intracellular calcium concentration in pulmonary artery smooth muscle [21,23].

The pathogenesis of group 3 PH involves chronic hypoxic pulmonary vasoconstriction and subsequent pulmonary vascular remodeling. The mechanism is not well understood, and its elucidation is important for establishing treatments for PH. Several factors have been reported to mediate vascular remodeling induced by hypoxic pulmonary vasoconstriction, as discussed below [24,25,26,27,28].

Hypoxic pulmonary vasoconstriction promotes the activation of innate immune responses and inflammation in microvessels via hypoxia-inducible factor 1α (HIF-1α) [16,29].

Elevated pulmonary arterial pressure due to chronic hypoxic pulmonary vasoconstriction increases vessel wall stress and strain and endothelial fluid shear stress. These mechanical cues promote vasoconstriction and vascular remodeling, which exacerbate PH [30]. Low shear stress, or changes in shear stress direction as seen in turbulence, promote endothelial proliferation and apoptosis, shape changes, vasoconstriction, coagulation, and secretion of substances that promote platelet aggregation [31,32]. Mechanosensors that receive the mechanical stimulus of vasoconstriction mediate vascular remodeling (Figure 1).

Cellular mechanosensors include mechanosensitive ion channels (e.g., stretch-activated channels), the cytoskeleton (e.g., actin filaments, intermediate filaments, and microtubules), and cell adhesion molecules (e.g., integrins). Among them, stretch-activated channels, which receive mechanical stimuli and convert them into biological signals, are among the candidate mechanosensors in pulmonary arteries, and the enhancement of Ca^2+^ entry through stretch-activated channels may contribute to pulmonary vascular remodeling after hypoxic pulmonary vasoconstriction. The increase in intracellular Ca^2+^ concentration due to the activation of Ca^2+^ channels plays an important role in fundamental cellular processes, such as the contraction, migration, and proliferation of pulmonary artery smooth muscle cells (PASMCs). This mechanotransduction via stretch-activated channels involves members such as transient receptor potential (TRP) and Piezo channels [33].

TRP channels constitute a superfamily of non-selective cation channels and are modulated by a variety of stimuli, including cold, heat, pH, membrane potential, mechanical stress (mechanical stimuli), neurohormones, and vasoactive substances.

Among the transient receptor potential canonical (TRPC) channels, TRPC1, TRPC3, TRPC4, TRPC5, and TRPC6 are sensitive to mechanical stimuli [34]. In preclinical studies using a 28-day mouse model of hypoxia-induced PH, intratracheal TRPC1 siRNA administration attenuated PH [35]. Furthermore, pulmonary vascular remodeling in response to hypoxia for 21 days was reduced in TRPC1-deficient mice (trpc1^−/−^ mice) [36]. These findings suggest that targeting TRPC 1 expression and function is a potential treatment for this disease. TRPC6 expression is also elevated in PASMCs from patients with idiopathic PAH and is involved in endothelin-mediated growth enhancement [24]. The increased expression of TRPC4 in human PAECs exposed to hypoxia was associated with capacitative Ca^2+^ entry via store-operated Ca^2+^ channels. This resulted in increased binding to activator protein-1 (AP-1), a transcription factor regulating the expression of genes involved in cell proliferation and migration, such as VEGF and PDGF genes [37]. A recent study has shown that elevated TRPC4 expression in pulmonary artery endothelial cells (PAECs) exacerbates hypoxia-induced PH by promoting apoptosis [25].

Among the transient receptor potential vanilloid (TRPV) channels, TRPV1, TRPV2, and TRPV4 are known to be mechanosensitive and are especially sensitive to osmotic stimuli [33]. TRPV1 is expressed in PAECs and PASMCs. This channel is Ca^2+^-permeable and contributes to many Ca^2+^ signaling pathways, including cell proliferation and cell migration associated with PH. Ca^2+^-mediated activation of TRPV1 channels in PAECs ameliorates endothelial cell inflammation, improves endothelium-dependent vasorelaxation, and protects animals from systemic hypertension through the endothelial nitric oxide synthase (eNOS)/nitric oxide pathway [37]. Upregulated TRPV1 had the opposite effect in PASMCs and could contribute to the pathogenesis of idiopathic PAH due to increased Ca^2+^ influx and excessive PASMC proliferation [37]. In PASMCs, TRPV1 activation is associated with changes in the cytoskeletal architecture, including the reorganization of F-actin, tubulin, and intermediate filament networks and the nuclear translocation of nuclear factor of activated T-cells (NFAT), and promotes PASMC migration [26]. In *trpv4*-deficient mice, chronic (4 weeks) hypoxia-induced right ventricular hypertrophy and pulmonary artery vascular remodeling via the muscularization of pulmonary arteries are attenuated [27].

Thus, an increase in intracellular calcium concentration through stretch-activated channels, which are mechanosensors, may play an important role, and novel therapies targeting TRP channels are awaited.

However, most of these experiments were performed in mice in which each molecule was knocked out from the whole body; since TRP channels are widely expressed in tissues throughout the body, the molecular entity and the point of action of the mechanosensor that triggers the development of PH remain unknown. Future studies using cell-specific knockout mice are needed to address these questions.

Alveolar hypoxia is a major determinant of the elevation of PVR and PAP in COPD patients. Accordingly, long-term oxygen therapy (LTOT) is a logical treatment for PH in COPD [28]. LTOT has long been the standard of care for hypoxemic COPD patients. In a Medical Research Council (MRC) study, LTOT patients had stable PAP after one year, whereas control patients had a significant increase in PAP [38]. However, the inability of LTOT to completely reverse the vasoconstriction and remodeling seen in patients with PH associated with COPD indicates that hypoxia-independent factors are also involved in the development of PH in COPD patients [16].

### 3.2. Cigarette Smoke

Cigarette smoke is a harmful substance that contributes to pulmonary vascular disease and is particularly associated with group 3 PH. Independent of airflow obstruction, smoking is associated with intimal hyperplasia [39], reduced expression of endothelial nitric oxide synthase [40], and inflammatory cell infiltration [41].

Cigarette smoke is also associated with mitochondrial fission and fusion imbalances, leading to mitochondrial oxidative stress and functional impairment in rat lung microvascular endothelial cells [42]. Cigarette smoke also causes aberrant mitophagy, increased mitochondrial oxidative stress, and reduced mitochondrial respiration. The inhibition of mitochondrial fission and mitochondria-specific antioxidants may be useful therapeutic strategies for cigarette smoke-induced endothelial injury and associated pulmonary diseases [42].

### 3.3. Decrease in Pulmonary Vascular Beds

In chronic respiratory diseases such as COPD and ILD, the normal vascular structure of the lungs is destroyed. The resulting reduction in pulmonary vascular beds is an important factor in the development and progression of group 3 PH [43].

The partial loss of pulmonary capillaries in group 3 PH compared to other PH subtypes explains the largely negative clinical trials of pulmonary vasodilators in group 3 PH [44].

A study by Bunel et al. based on the morphological analysis of the pulmonary vasculature of lung explants from patients who underwent lung transplantation for COPD showed that the morphological correlates or characteristics of severe PH-COPD were substantially increased muscularization of the pulmonary microvasculature (arterioles and/or venules) and low capillary density when compared to those in moderate PH-COPD [45]. In addition, there were no significant differences in muscular-type pulmonary arteries that are relevant in PAH (group 1), and typical PAH lesions, such as plexiform and onionskin lesions (concentric laminar intimal fibrosis), were not detected, highlighting that at least morphological phenotypes differ between severe PH-COPD and PAH.

### 3.4. Hypoxia-Inducible Factors (HIFs)

HIFs are transcription factors that respond to decreases in available oxygen in the cellular environment, i.e., hypoxia. HIF signaling plays a pivotal role in the onset and pathogenesis of PH [46]. Prolonged exposure to hypoxia in pulmonary vascular and inflammatory cells causes the HIF isoform to transcriptionally activate a series of genes that regulate vascular tone, angiogenesis, metabolism, and proliferation, leading to PAEC dysfunction, PASMC proliferation, inflammation, and oxidative stress, followed by pulmonary vascular remodeling [16,29,46]. The expression of HIF-1α, vascular endothelial growth factor (VEGF)—a potent regulator of vascular permeability—and VEGF receptors was higher in COPD patients who were smokers [47]. This results in the activation of the inflammatory transcription factor nuclear factor-κB (NF-κB) [16]. HIFs in infiltrating macrophages in hypoxia-induced PH promote the release of cytokines through crosstalk with the NF-kB signaling pathway. This enhances macrophage recruitment, chemotaxis, and polarization. Furthermore, in response to hypoxia, mitochondrial oxidative phosphorylation is inhibited while glycolysis is enhanced. This enhances the production of ROS, which in turn activates and stabilizes HIFs [48].

Interestingly, along with chronic hypoxia, other factors that contribute to the development of PH (gene variants, vasoconstriction, endothelial dysfunction, mitochondrial abnormalities, dysregulated cell growth, and inflammation) enhance the activation of the HIF signaling pathway [46].

Global, inducible, and cell-specific deletion of HIF isoforms and HIF pathway molecules indicate the cell-type- and context-specific roles of HIF isoforms in the early and late stages of PH development in adult mice [46]. Pharmacologically desirable effects of HIF inhibition have been investigated as a treatment for various cancers, retinal neovascularization, and PH [49].

### 3.5. Endothelial Dysfunction

Factors contributing to endothelial dysfunction, such as increased endothelin-1, decreased prostacyclin, and decreased nitric oxide, are observed in COPD and ILD [16,17]. However, drugs targeting these three pathways are not approved for patients with COPD-associated PH due to lack of evidence [50].

#### 3.5.1. Endothelin-1

Endothelin-1 is the most potent pulmonary vasoconstrictor and is a co-mitogen for SMCs and fibroblasts. Not only hypoxia but also cigarette smoke extract can independently induce endothelin-1 expression in PAECs [16]. Endothelin-1 in peripheral blood is elevated in patients with COPD and ILD [51,52]. Stolz et al. reported that the oral administration of the endothelin receptor antagonist bosentan not only failed to improve exercise capacity but also deteriorated hypoxaemia and functional status in severe chronic obstructive pulmonary disease patients without severe pulmonary hypertension at rest [53]. In contrast, Valerio et al. reported bosentan treatment resulted in a significant improvement of PAP, PVR and 6-min walk distance (6MWD) [54]. Thus, there are still insufficient data to fully support the effects of endothelin receptor antagonists (ERAs) on pulmonary hemodynamics and exercise tolerance in patients with COPD-PH.

#### 3.5.2. Nitric Oxide

Endothelial nitric oxide synthase expression in the intima of pulmonary arteries is reduced in smokers [40]. Hypoxia has been shown to reduce eNOS expression and nitric oxide production in the lungs of newborn piglets exposed to chronic hypoxia [55]. A phase III randomized, double-blind, placebo-controlled trial (REBUILD trial) revealed that there was no demonstrable benefit from inhaled nitric oxide in patients with fibrotic interstitial lung disease receiving supplemental oxygen in daily physical activity assessed by means of actigraphy (NCT03267108) [56].

#### 3.5.3. Prostacyclin (PGI_2_)

The expression of prostacyclin (PGI_2_) synthase is lower in the endothelium of human emphysema lung tissue than in the endothelium of normal lungs [57]. Cigarette smoke extract can reduce prostacyclin (PGI_2_) synthase expression in PAECs [16,57]. Pulmonary arterial hypertension of the newborn is a syndrome caused by chronic hypoxia, characterized by decreased vasodilator function, a marked vasoconstrictor activity, and proliferation of smooth muscle cells in the pulmonary circulation. Antioxidant therapy with melatonin induces prostanoid expression in neonates [58].

#### 3.5.4. ErbB3

Human epidermal growth factor receptor 3 (ErbB3), a member of the ErbB family of receptor tyrosine kinases, showed significantly elevated expression levels in serum, lungs, distal pulmonary arteries, and PAECs isolated from patients with hypoxia-induced PH compared with those from healthy donors [59]. ErbB3 expression levels in serum were significantly correlated with mPAP, PVR, or the cardiac index. ErbB3 silencing successfully reduced the proliferation of PH-PAECs, whereas ErbB3 overexpression promoted tube formation of PAECs and altered the cytoskeletal structure, indicating that ErbB3 expression is important for the dysfunction of PAECs in PH. The overexpression of ErbB3 induced via the injection of adeno-associated virus vectors stimulated hypoxia-induced endothelial cell proliferation, exacerbated pulmonary artery remodeling, increased right ventricular systolic pressure, and promoted right ventricular hypertrophy in a mouse model of PH. Conversely, the systemic deletion or endothelial cell-specific knockout of ErbB3 had the exact opposite effect. ErbB3 serves as a novel biomarker and a promising target for the treatment of hypoxia-induced PH.

### 3.6. Inflammation

The presence of inflammation in hypoxic PH has been histologically found around pulmonary vascular lesions, with the inflammatory infiltrate comprising macrophages, mast cells, neutrophils, platelets, T lymphocytes, B lymphocytes, and dendritic cells [48]. Perivascular inflammation significantly correlates with the intima and intimal fractional thickness, increases pulmonary artery pressure, and contributes to the structural remodeling of pulmonary vascular vessels [48,60].

Pulmonary artery remodeling in mice with hypoxia-induced PAH progressively worsens over time, with the infiltration of CD68 macrophages being predominant in the early stages and pulmonary vascular remodeling being predominant in the later stages [61]. The assessment of the proportions of M1 and M2 macrophages using flow cytometry showed the predominance of M1 macrophages with an inflammatory phenotype in the early stages of hypoxia exposure and a decrease in the later stages [61].

Various inflammatory cytokines have been reported to be involved in the development of vascular remodeling in PAH, and interleukin (IL)-6 in particular is known to play an important role in its pathogenesis [62]. In the validation of 148 COPD cases, serum monocyte chemoattractant protein-1 (MCP-1) and IL-1beta levels in cases of PH-associated COPD with mPAP ≥ 25 mmHg were not different from the levels in the no-PH group, while serum IL-6 levels were significantly elevated in the PH group. Furthermore, serum IL-6 levels were positively correlated with mPAP [63]. These results suggest that IL-6 may be involved in the progression of PH not only in cases of PAH but also in cases of group 3 PH including COPD.

### 3.7. Oxidative Stress

Increased oxidative stress in PH is often caused by chronic or persistent hypoxia [64]. Important sources of ROS in PH are the mitochondrial electron transport chain and the NADPH oxidase (NOX) family of enzymes, which produce superoxide by transferring electrons from NADPH to oxygen. This elevated production of ROS contributes to pulmonary vascular remodeling, endothelial dysfunction, altered vasoconstrictive responses, inflammation, and changes in the extracellular matrix [64,65].

Antioxidant treatments have been tested in experimental animal models. For example, Huang et al. reported that melatonin treatment significantly attenuated the levels of right ventricular systolic pressure and oxidative and inflammatory markers in hypoxic animals with a marked increase in eNOS phosphorylation in the lungs [66]. Gonzalez-Candia et al. reported that melatonin had long-lasting beneficial effects on pulmonary vascular reactivity and redox balance in chronic hypoxia in newborn lambs gestated and born at 3600 m [67].

Elevated levels of ROS induce lipid peroxidation, generating reactive lipid dicarbonyls, including isolevuglandins (IsoLGs) (also called isoketals), 4-oxo-nonenal, and malondialdehyde, and α,β-unsaturated aldehydes, such as acrolein and 4-hydroxy-nonenal [68,69]. IsoLGs are associated with various cardiovascular diseases, such as hypertension, atherosclerosis, atrial fibrillation, and heart failure [70,71,72,73]. 2-Hydroxybenzylamine (2-HOBA) is a natural product found in buckwheat seeds and acts as an antioxidant and scavenger of IsoLGs [74,75]. Egnatchik et al. reported that treatment with 2-HOBA prevented the development of PAH in BMPR2 mutant mice [76]. The phase II clinical trial “Clinical Trial of 2-HOBA in Pulmonary Arterial Hypertension” is investigating its safety and efficacy in patients with PH (NCT06176118).

### 3.8. Genetic Predisposition

Eddahibi et al. reported that serotonin (5-hydroxytryptamine, 5-HT) and its transporter (5-HTT) play an important role in pulmonary vascular smooth muscle hyperplasia and vascular remodeling associated with experimental hypoxic PH and human idiopathic PAH [77,78]. The overexpression of 5-HTT seen in the pulmonary vessels of patients with idiopathic PAH is at least partly related to an insertion/deletion polymorphism in the promoter region of the 5-HTT gene with long (L) and short (S) forms: the L allele drives a 2- to 3-fold higher rate of 5-HTT gene transcription than does the S allele. In 103 COPD cases, a search for gene polymorphisms in the promoter region of 5-HTT revealed that the severity of PH, as assessed by examining the pulmonary artery pressure, was higher in cases with the LL genotype than in those with the LS/SS genotypes [79].

Swietlik et al. reported that heterozygous, high-impact, likely loss-of-function variants in the kinase insert domain receptor (KDR) gene, which encodes vascular endothelial growth factor receptor 2 (VEGFR2), were strongly associated with significantly reduced transfer coefficient for carbon monoxide (KCO), older age at diagnosis, and parenchymal abnormalities assessed by means of computed tomography [80]. Eyries et al. reported two index cases with KDR mutations after prospectively screening a series of 311 unrelated patients referred for PAH genetic investigation; the KDR mutations were associated with a particular form of PAH characterized by a low diffusing capacity for carbon monoxide adjusted for hemoglobin (D (LCO)c) and radiological evidence of parenchymal lung disease, including interstitial lung disease and emphysema [81].

In neonates, variants of endothelin-1 (EDN1), carbamoyl phosphate synthetase I (CPS1), neurogenic locus notch homolog protein 3 (NOTCH3), and SMAD family member 9 (SMAD9) are associated with the risk of developing persistent PH of the newborn (PPHN) [82,83].

### 3.9. Comorbidities

Comorbidities impact a large proportion of patients with COPD and ILD. Comorbidities include allergic disease, coronary heart disease, congestive heart failure, diabetes, metabolic syndrome, and sleep apnea, which affect pulmonary hypertension and its prognosis [84,85].

## 4. Consequential Factors of Pathogenesis of Group 3 PH

### 4.1. Pulmonary Vascular Remodeling

Vascular remodeling due to hypoxia is often restricted to the media. The analysis of pulmonary vessels in COPD patients with PH demonstrated marked intimal thickening, medial hypertrophy, and muscularization of small arterioles [86]. Endothelial cell dysfunction and the proliferation of smooth muscle cells (SMCs) and fibroblasts contribute to pulmonary vascular remodeling [44]. These pulmonary vascular changes occurred in normoxic patients with mild airflow obstruction and in smokers without substantial impairment, suggesting that vascular remodeling may be driven by hypoxia and by mechanisms independent of hypoxia [39].

Consistent with these findings in humans, studies in mice exposed to cigarette smoke have shown that pulmonary vascular remodeling and PH precede the development of emphysema and are independent of hypoxia [86]. Furthermore, the combination of smoking and hypoxia has a synergistic effect on the vasculature. Experiments in guinea pigs exposed to smoking and hypoxia showed a more pronounced increase in mPAP and remodeling of small blood vessels compared to those in guinea pigs exposed to a single stimulus [87].

In COPD and ILD, vascular remodeling is known to be caused not only by alveolar hypoxia but also by factors contributing to endothelial dysfunction, such as increased endothelin-1, decreased prostacyclin and nitric oxide, increased oxidative stress, inflammation, and genetic predisposition [16,17].

When pulmonary hypertension is induced by chronic hypoxia (normobaric), remodeling occurs in all three layers of the pulmonary vasculature—the intima, tunica media, and adventitia—and, in particular, the number of adventitial fibroblasts increases [88]. Considerable experimental evidence supports the possibility that fibroblasts play an important role in the vascular response under hypoxic conditions [88,89]. Fibroblasts appear to have a unique ability to proliferate, transdifferentiate, and migrate under hypoxic conditions [89]. The proliferative response to hypoxia is dependent on the activation of Gαi and Gq kinase family members and the subsequent stimulation of protein kinase C and mitogen-activated protein kinase family members. Hypoxia, acting through the Gα(iota)- coupled pathway, can also directly upregulate the expression of α-smooth muscle actin in fibroblast subpopulations. This suggests that hypoxia may play a direct role in mediating fibroblast-to-myofibroblast “transdifferentiation” in the vascular wall.

Elevated oxidative stress is a pivotal pathological feature in PH, often caused by chronic hypoxia or sustained low-oxygen conditions [65]. The increase in reactive oxygen species (ROS) is associated with a higher expression of NADPH oxidases in the pulmonary arteries and contributes to vascular remodeling by promoting smooth muscle cell proliferation and fibrosis [65].

Hypoxic stress also induces an increase in the number of monocyte progenitor cells from the bone marrow, which may upregulate the expression of receptors for chemokines produced in the pulmonary circulation and promote their specific recruitment to the lung sites [90]. Macrophages/fibrocytes exert a paracrine effect on pulmonary vascular wall cells, stimulating fibroblast and SMC proliferation, phenotypic changes, and migration [90].

### 4.2. Endothelial-to-Mesenchymal Transition (EndMT)

Endothelial-to-mesenchymal transition (EndMT) is a complex biological process in which endothelial cells undergo a series of molecular events that change them to the mesenchymal phenotype (myofibroblasts, SMCs, etc.) (Figure 2). EndMT is involved in several cardiovascular diseases, including atherosclerosis, pulmonary hypertension, and valvular disease [91]. Ranchoux et al. reported that EndMT occurred in situ in human PAH as well as in experimental PH induced by monocrotaline or by combined exposure to hypoxia and vascular endothelial growth factor receptor blockade. EndMT was shown to be associated with altered bone morphogenetic protein receptor 2 (BMPR2) signaling and to be involved in occlusive vascular remodeling in PAH [92]. EndMT is also associated with factors that contribute to PAH pathogenesis, including the hypoxic response, inflammation, oxidative stress, and redox signaling [93,94,95].

Bhattarai et al. evaluated resected lung tissue with immunohistochemical staining and reported that EndMT causes mesenchymal traits in lung endothelial cells from smokers with normal lung function, small airway disease, and moderate COPD. A decrease in junctional CD31^+^ endothelial cells was observed in the intimal layer in all smoking groups compared to nonsmoker controls. Increases in the mesenchymal markers N-cadherin and vimentin were also observed in all smoking groups [96].

Gaikwad et al. reported increased expression levels of the mesenchymal biomarkers N-cadherin, S100A4, and vimentin in the pulmonary arterial layers of IPF patients compared to those of normal controls, indicating that resident cells may be actively transitioning to mesenchymal traits [97]. Significant expression of TGF-β1, pSmad-2/3, Smad-7, and β-catenin was evident across all arterial sizes in IPF. The expression rates of intimal TGF-β1 and β-catenin were strongly correlated with those of intimal vimentin and intimal N-cadherin [98].

### 4.3. Mitochondrial Dysfunction

Mitochondria are organelles in eukaryotic cells that function as cellular powerhouses, dependent on the continuous availability of oxygen. Under hypoxic conditions, metabolic abnormalities disrupt the steady state of the mitochondrial network, causing mitochondrial dysfunction and generating large amounts of ROS that can further damage cells [99]. Mitochondrial ROS (mtROS) plays a significant role in the development of chronic hypoxia-induced PH [65,100,101]. The administration of MitoQ, an antioxidant that targets mtROS, was able to alleviate symptoms of PH, as evidenced by a reduction in right ventricular pressure and pulmonary arterial remodeling in chronic hypoxia-induced PH [100]. 

Bhansali et al. reported an excessive rise in mtROS production and disrupted membrane potential, accompanied by enhanced DNA damage and reduced autophagy was observed, highlighting the ‘apoptosis resistance’ phenotype, in hypoxic PASMCs [102]. 

As far as energy metabolism is concerned, the exact effects of chronic hypoxia on mitochondrial energy metabolism in the lungs are not yet fully understood. Further studies are needed to clarify this point.

Given that pulmonary arterial remodeling can be caused by the excessive proliferation and apoptosis resistance of PASMCs, the inhibition of cell proliferation or induction of cell apoptosis is considered an efficient therapeutic strategy for PH. For example, Li et al. reported that hypoxia promoted the proliferation of human PASMCs, inhibited the activity of caspase-3, and increased ROS levels, mitochondrial membrane potential, and the expression of HIF-1alpha and PDK4, which induced glycolysis [103]. Sirtuin 6 overexpression could inhibit the proliferation of human PASMCs, as well as increase the apoptosis rate and reduce the levels of ROS, the mitochondrial membrane potential, and the expression of HIF-1alpha and PDK4 in hypoxic human PASMCs [103].

### 4.4. MicroRNA (miRNA)

MicroRNAs (miRNAs) are small, single-stranded, non-coding RNA molecules of 21 to 23 nucleotides and are involved in RNA silencing and the post-transcriptional regulation of gene expression. miRNAs are related to the pathogenesis of hypoxia-induced PH (Table 2).

Hypoxia-inducible microRNA-210 (miR-210) has been implicated as a causative factor promoting several PH subtypes, including PAH (group 1 PH) and PH due to hypoxic lung disease (group 3 PH) [106,110]. The endogenous blood-borne transport of miR-210 to pulmonary vascular endothelial cells was shown to promote the development of PH in an in vivo study [110]. These results may facilitate the development of blood-based miR-210 technology for the diagnosis and treatment of PH.

In contrast, MicroRNA-212-5p, an anti-proliferative miRNA, attenuates hypoxia and sugen/hypoxia-induced PH in rodents [108]. The levels of miR-212-5p are upregulated in PASMCs and lungs in human PH patients and rodent models of PH. SMC-specific knockout of miR-212-5p significantly exacerbated hypoxia-induced pulmonary vessel wall thickening and PH in mice. The administration of exogenous miR-212-5p attenuates both chronic hypoxia-induced PH in mice and sugen/hypoxia-induced severe PH in rats. Jin et al. reported that blood hsa_circNFXL1_009 levels were reduced in COPD patients with PH and that the administration of exogenous hsa_circNFXL1_009 attenuated the hypoxia-induced proliferation, apoptotic resistance, and migration of human PASMCs [111].

Zhou et al. reported that hsa_circ_0016070 expression was significantly increased in the lungs of COPD patients with PH and that hsa_circ_0016070 promoted PASMC proliferation by regulating miR-942/CCND1 [112].

## 5. Right Heart Failure

Right heart failure is a clinical sign characterized by right ventricular dysfunction, resulting in inadequate blood flow and elevated filling pressures. In PH, an adapted right ventricle is slightly dilated, maintains stroke volume and contractility, and shows normal filling pressures. On the other hand, a right ventricle that is not adapted to PH is dilated, has reduced stroke volume and contractility, and shows an increase in filling pressure.

Right ventricular function is an important prognostic factor in PH [113]. Right heart failure in PH is the result of increased afterload. The function of the cardiopulmonary system is influenced not only by right ventriculo-pulmonary artery coupling but also by factors such as ventilation–perfusion matching, interventricular and interatrial interactions, right ventricular perfusion, metabolism, and myocardial oxygen consumption (Figure 3).

The right ventricle in chronic lung disease and hypoxia undergoes remodeling with increased dimensions and hypertrophic changes. Right heart failure due to these changes is known as “cor pulmonale” [9,114,115]. Hilde et al. reported that impaired right ventricular (RV) systolic function, hypertrophy, and dilation were present even at a slightly increased mPAP, which indicates an early impact on RV function and structure in patients with COPD [116]. The majority of these patients did not have PH, highlighting that even a slight increase in mPAP has an important impact on RV function and structure. However, the lack of association with PH in the early stages of COPD also suggests other mechanisms.

In group 3 PH, even after adjustment for RV afterload, male sex was associated with RV dysfunction, as assessed by examining the right ventricular fractional area change (RVFAC) [117]. RV dysfunction was associated with an increased risk of heart failure hospitalization or death. RV dysfunction identifies group 3 PH patients who are at risk of a poor prognosis.

As for cardiac function at high altitudes, healthy volunteers had decreased left ventricular (LV) mass (adjusted for changes in body surface area) and slightly reduced LV diastolic function after trekking to Mt Everest Base Camp (5300 m) [118]. There were no changes in the LV or RV ejection fraction. Lowland residents arriving at a high altitude (3750 m) experienced increased mPAP (20–25 mmHg) and changes in RV and LV diastolic function, as assessed by means of echocardiography, while RV systolic function was maintained [119]. After 10 days of altitude acclimatization, mPAP (measured at 4850 m altitude) increased slightly (26 mmHg), but cardiac function did not change further. These results show that healthy individuals exposed to mild hypoxia-induced PH maintain systolic function despite a slight impairment of ventricular filling mechanisms [120].

The primary factor leading to right heart failure in PH is increased afterload. Indeed, severe RV dysfunction and dilation were observed in patients with end-stage PAH despite concomitant PAH medication, whereas RV function and morphology improved when the RV pressure load was reduced by means of lung transplantation [121]. In group 3 PH, hypoxia and increased right ventricular afterload may influence the development of right heart failure. Chronic hypoxia exposure, sugen/hypoxia, tobacco smoke exposure, and bleomycin treatment were studied using a group 3 PH animal model [122]. Future studies on right heart failure due to group 3PH are expected.

## 6. Clinical Implications

Data supporting the use of PAH therapy in group 3 PH are limited, and further studies are required. Vasoactive medications may be effective in patients with severe hemodynamic parenchymal lung disease, and these patients may be a target population for future studies.

The INCREASE trial (NCT02630316) evaluated the safety and efficacy of inhaled treprostinil, an analog of prostacyclin, in patients with PH due to ILD [123]. In this patient population, inhaled treprostinil improved exercise capacity from baseline in the 6-min walk test as compared with placebo. In addition, treatment with inhaled treprostinil was associated with a lower risk of clinical worsening, a reduction in NT-proBNP levels, and fewer exacerbations of the underlying lung disease compared with placebo. The PERFECT study (NCT03496623) evaluated the safety and efficacy of inhaled treprostinil in PH associated with COPD [124]. The results showed that numerically more adverse events, deaths, treatment discontinuations, and study discontinuations occurred in patients receiving inhaled treprostinil compared to placebo. In addition, patients receiving inhaled treprostinil did not show an improvement in 6MWD compared to placebo. Overall, the trial showed that the risks in treating PH-COPD patients with inhaled treprostinil outweighed the potential benefits, thereby justifying early discontinuation. Thus, the efficacy and safety of the same drug varied depending on the type of parenchymal lung disease. These studies suggest that future clinical trials for group 3 PH will require disease-specific and fine-tuned designs. It is hoped that future clinical studies targeting the small molecules described in this review will draw on the lessons learned from these studies.

Table 3 shows ongoing trials evaluating the effects of various PH therapies in patients with PH associated with COPD. Changes in mitochondrial metabolic parameters, as assessed using the Agilent Seahorse extracellular flux bioanalyzer, will be evaluated after pioglitazone administration (NCT0633679). We look forward to the future development of therapies targeting other molecular biological mechanisms discussed in this review.

For group 1 PAH, a new treatment with sotatercept, a fusion protein that traps activins and growth differentiation factors, has begun [125]. The treatment of group 3 PH with selective pulmonary vasodilators, such as a PGI_2_ analog, is currently being attempted [123]. In the future, in addition to selective pulmonary vasodilators, we hope to develop treatments for group 3 PH that target novel molecules that improve vascular remodeling (Figure 4).

## 7. Conclusions

Elucidating the pathogenesis of PH associated with chronic lung disease and hypoxia is essential for the development of new treatments for this disease. In this review, we discussed the mechanisms of group 3 PH, focusing on pulmonary vasoconstriction and vascular remodeling. Several factors have been reported to mediate vascular remodeling induced by hypoxic pulmonary vasoconstriction, such as HIF-1α and mechanosensors, including TRP channels. Current efforts are focused on new target molecules, such as mechanoreceptors that sense pulmonary vasoconstriction. These could lead to new therapies to inhibit vascular remodeling in the future.

## Figures and Tables

**Figure 1 ijms-26-00835-f001:**
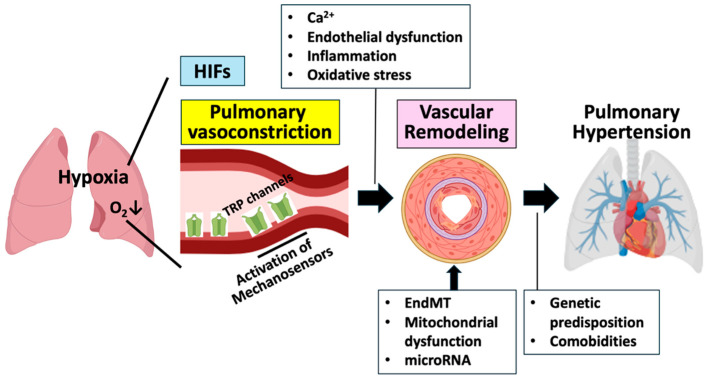
Vascular remodeling following mechanosensor-mediated hypoxic pulmonary vasoconstriction. Hypoxia induces pulmonary vasoconstriction. Mechanosensors (e.g., stretch-activated channels) receive the mechanical stimulus of vasoconstriction and mediate vascular remodeling and pulmonary hypertension. Downward arrow means decrease. EndMT, endothelial-to-mesenchymal transition; HIFs, hypoxia-inducible factors; TRP channels, transient receptor potential channels.

**Figure 2 ijms-26-00835-f002:**
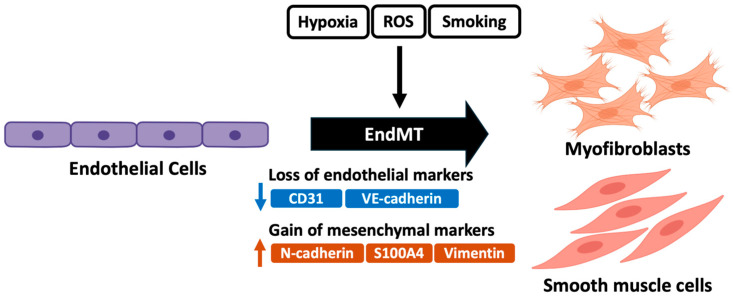
The induction of endothelial-to-mesenchymal transition (EndMT) in group 3 PH. Endothelial cells undergo a series of molecular events that change them to the mesenchymal phenotype, such as myofibroblasts and smooth muscle cells. ROS, reactive oxygen species; S100A4, Protein S100-A4; N-cadherin, neural cadherin; VE-cadherin, vascular endothelial cadherin.

**Figure 3 ijms-26-00835-f003:**
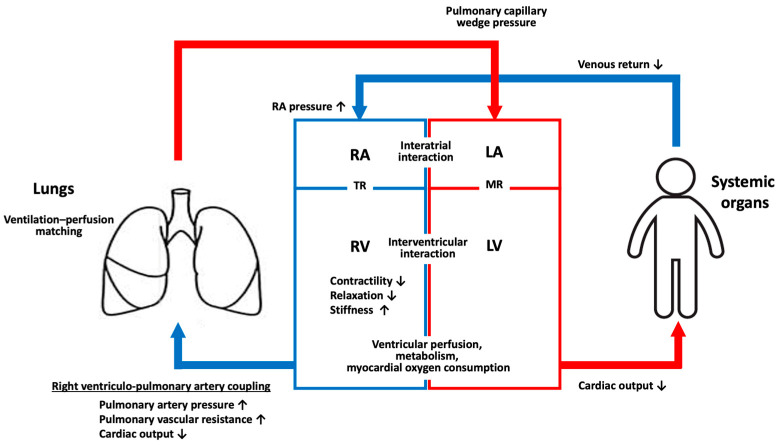
Components of the cardiopulmonary system. Blue lines represent venous blood and red lines represent arterial blood. The upward arrows indicate an increase and the downward arrows indicate a decrease. LA, left atrium; LV, left ventricle; MR, mitral regurgitation; RA, right atrium; RV, right ventricle; TR, tricuspid regurgitation.

**Figure 4 ijms-26-00835-f004:**
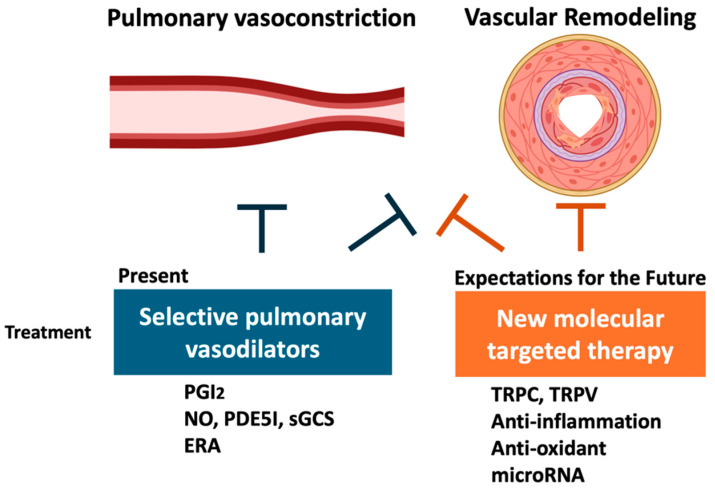
Treatments from selective pulmonary vasodilator therapy to novel molecularly targeted therapy. In the future, we hope to develop a group 3 PH therapy targeting novel molecules that improve vascular remodeling, in addition to selective pulmonary vasodilators. ERA, endothelin receptor antagonists; NO, nitric oxide; PDE5I, phosphodiesterase 5 inhibitor; PGI_2_, prostacyclin; sGCS, soluble guanylate cyclase stimulator; TRPC, transient receptor potential canonical; TRPV, transient receptor potential vanilloid.

**Table 1 ijms-26-00835-t001:** Pathogenesis of group 3 pulmonary hypertension and/or hypoxia.

Causal Factors	Consequential Factors
Hypoxic pulmonary vasoconstriction	Pulmonary vascular remodeling
Cigarette smoke	Endothelial-to-mesenchymal transition (EndMT)
Decrease in pulmonary vascular beds Hypoxia-inducible factors (HIFs)	Mitochondrial dysfunction
Endothelial dysfunction	MicroRNA (miRNA)
Increase in endothelin-1	
Reduction in endothelial nitric oxide synthase (eNOS)	
Reduction in prostacyclin (PGI_2_) synthase	
Inflammation Oxidative stress	
Genetic predisposition Comorbidities	

**Table 2 ijms-26-00835-t002:** MicroRNAs related to pathogenesis of hypoxia-induced pulmonary hypertension.

MicroRNA	Role	Expression	Target	Effects	References
miR-17	facilitative	increased	BMPR2	Increased PASMC proliferation	Pullamsetti et al. [104]
miR-30a	facilitative	increased	p53	Inhibition of microRNA-30a alleviates vascular remodeling in Su5416/hypoxia-induced PH animals	Ma et al. [105]
miR-210	facilitative	increased	ISCU1/2	Iron–sulfur deficiencies and promotion of PH	White et al. [106]
miR182	suppressive	-	Myadm	miR-182 gain-of-function significantly inhibited the pathological progression in hypoxia-induced PH	Bai et al. [107]
miR-212-5p	suppressive	increased		Suppression of PASMC proliferation in hypoxia-induced PH in rodents	Chen et al. [108]
miR-320-3p	suppressive	decreased	KLF5, HIF-1α	Inhibition of proliferation and migration and promotion of apoptosis in hypoxic PASMCs	Ding et al. [109]

BMPR2, bone morphogenetic protein receptor 2; HIF-1α, hypoxia-inducible factor-1α; ISCU1/2, iron–sulfur cluster assembly proteins 1/2; KLF5, Krüppel-like factor 5; PASMC, pulmonary artery smooth muscle cell; PH, pulmonary hypertension.

**Table 3 ijms-26-00835-t003:** Ongoing trials evaluating the effects of various therapies in patients with PH associated with COPD.

Study Title	NCT Number	Interventions	Primary Outcome
MK-5475-013 INSIGNIA-PH-COPD: A Study of the Efficacy and Safety of MK-5475 (an Inhaled sGC Stimulator) in Adults With PH-COPD	NCT05612035	MK-5475	Mean change from baseline in 6-min walk distance
A Mechanistic Study of Inhaled Nitric Oxide in COPD	NCT05785195	Inhaled NO	Change in ventilation–perfusion ratio (V/Q) from baseline
Long-term Oxygen Therapy in Patients with Chronic Obstructive Pulmonary Disease Who Live at High Altitude	NCT03020212	Oxygen	Development of PH, assessed via echocardiogram
Hyperoxia During Pulmonary Rehabilitation in Chronic Lung Disease—Does it Matter?	NCT06174207	Oxygen	Constant Work Rate Exercise Test
Bioenergetic Effect of Pioglitazone in CLD-PH	NCT06336798	Pioglitazone	Change in mitochondrial metabolism parameters
COPD Exacerbation and Pulmonary Hypertension Trial	NCT04538976	Sildenafil	Time alive and out of hospital
Tadalafil for Severe Pulmonary Hypertension Due to Chronic Obstructive Pulmonary Disease	NCT05844462	Tadalafil	6-min walk
Breathe Easier with Tadalafil Therapy for Dyspnea in COPD-PH	NCT05937854	Tadalafil	Severity of patient-reported dyspnea
Tadalafil for Severe Pulmonary Hypertension Due to Chronic Obstructive Pulmonary Disease	NCT05844462	Tadalafil	6-min walk

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
