# Peer review of "Pathophysiology of Group 3 Pulmonary Hypertension Associated with Lung Diseases and/or Hypoxia"

_ijms, 2025, doi:10.3390/ijms26020835_

Round 1

Reviewer 1 Report (New Reviewer)

Comments and Suggestions for Authors

ABSTRACT

1.     English should be improved.

2.     The Hypoxic Pulmonary Vasoconstriction (HPV) abbreviation should be changed as HPV is known for human papillomavirus and can cause confusion to the readers.

INTRODUCTION

1.     Pulmonary hypertension is estimated at more than 1% of the global population. Authors should update their references with the most new records from 2023/2024.

2.     All references must be updated 

3.     The introduction must be reorganized starting from the daisies definition cause to the treatment and then the specific aims of the study following the review known structure.

REVIE

1.     The authors didn’t clearly present the different sections and there is no coherence between the different sub-paragraphs.

2.      Provide full abbreviations of NFAT(line 160), KO(line 170), PVR( line 172), PAP(line 173)

3.     Figure 3 can be improved. Figure should be clear and more elaborative that can provide an easy and clear understanding of the topic. 

4.     In Line 415, I believe the term Cor Pulmonale has been wrongly used. It is when right heart fails due to changes. Only changes are not called cor pulmonale.

5.     Writing must be improved very poor grammar and English.

CONCLUSION

1.     The conclusion is average and can be improved.

2.     In line 491, I believe hyperoxia is wrongly used. Hypoxia is the right terminology.

Comments on the Quality of English Language

The English of the manuscript is poor and must be improved I suggest that authors send their manuscript to a native English speaker to review the grammar or use the scientific English editing services to improve this version.

Author Response

Response to reviewer 1

We greatly appreciate the reviewer’s comments.

ABSTRACT

Comment: 1. English should be improved.

Response: Thank you for your comment. In accordance with the reviewer’s comment, our manuscript has undergone English language editing by MDPI.

Comment: 2. The Hypoxic Pulmonary Vasoconstriction (HPV) abbreviation should be changed as HPV is known for human papillomavirus and can cause confusion to the readers.

Response: In accordance with the reviewer’s comment, we have removed (HPV) abbreviation from the manuscript.

INTRODUCTION

Comment: 1. Pulmonary hypertension is estimated at more than 1% of the global population. Authors should update their references with the most new records from 2023/2024.

Response: Thank you for your comment. In accordance with the reviewer’s comment, we have added the following sentences to Introduction: “Pulmonary hypertension (PH) is a global health problem. Current estimates put the prevalence of PH at least 1% of the world's population, which increases up to 10% in individuals aged more than 65 years [1, 2].” (lines 33-35).

Comment: 2. All references must be updated.

Response: In accordance with the reviewer’s comment, we have updated references and added 16 new references.

Comment: 3. The introduction must be reorganized starting from the daisies definition cause to the treatment and then the specific aims of the study following the review known structure.

Response: In accordance with the reviewer’s comment, we have ordered the Introduction in the following order: “1.1. Definition”, “1.2. Clinical classification”, and “1.3. Aims of this review”.

REVIE

Comment: 1. The authors didn’t clearly present the different sections and there is no coherence between the different sub-paragraphs.

Response: In accordance with the reviewer's comments, we have restructured the sections and paragraphs according to the order in Table 1.

Comment: 2. Provide full abbreviations of NFAT(line 160), KO(line 170), PVR( line 172), PAP(line 173).

Response: In accordance with the reviewer's comments, we have provided full abbreviations of NFAT (line 190), knockout (line 201), PVR( line 125),and PAP(line 57).

Comment: 3. Figure 3 can be improved. Figure should be clear and more elaborative that can provide an easy and clear understanding of the topic. 

Response: In accordance with the reviewer's comments, we have revised Figure 3.

Comment: 4. In Line 415, I believe the term Cor Pulmonale has been wrongly used. It is when right heart fails due to changes. Only changes are not called cor pulmonale.

Response: In accordance with the reviewer’s comment, we have added the following sentences: “Right heart failure due to these changes is known as "cor pulmonale" [9, 105, 106].” (lines 516-517).

Comment: 5. Writing must be improved very poor grammar and English.

Response: In accordance with the reviewers' comments, our manuscript was edited in English by MDPI.

CONCLUSION

Comment:1. The conclusion is average and can be improved.

Response: In accordance with the reviewer’s comment, we have added the following sentences to Conclusions: “Several factors have been reported to mediate vascular remodeling induced by hypoxic pulmonary vasoconstriction, such as HIF-1α and mechanosensors, including TRP channels. Current efforts are focused on new target molecules, such as mechanoreceptors that sense pulmonary vasoconstriction. These could lead to new therapies to inhibit vascular remodeling in the future.” (lines 598-602).

Comment:2.  In line 491, I believe hyperoxia is wrongly used. Hypoxia is the right terminology.

Response: Thank you for your comment.  In accordance with the reviewer’s comment, the term ‘hyperoxia’ was amended to ‘hypoxia’ (line 595).

Reviewer 2 Report (New Reviewer)

Comments and Suggestions for Authors

Overall, the review addresses the pathophysiology of group 3 pulmonary hypertension. However, the paper lacks precision in the concepts it aims to present. It is necessary to review the lack of references for some statements.

1. Introduction:
This section discusses pulmonary hypertension; however, it is essential to include the clinical classification of pulmonary hypertension mentioned in section 2.2.

2. Pulmonary Hypertension:
The definitions of hypertension and COPD should be included in the introduction section.

3. Pathogenesis of Group 3 Pulmonary Hypertension:
Table 1: The table suggests a causal relationship between the pairs listed. For example: inflammation and miRNAs. The table needs improvement.
It is recommended to organize the sections based on causes and consequences. For instance, section 3 could represent causes and section 4 consequences.

3.1:
The type of injury related to hypoxia (chronic, acute, or intermittent) should be specified. Additionally, detailed explanations of the translational animal models used, including the duration of hypoxia exposure, are necessary.
While vascular remodeling is mentioned as a consequence of HAP, this occurs after prolonged injury periods. Moreover, components of the immune and inflammatory systems via HIF-1α are mentioned without detailing the molecular mechanisms involved.

  • Line 112: Missing reference.
  • Line 119: Missing reference.
  • Figure 1: This figure does not contribute to the review. It would be more appropriate to illustrate the relationship between hypoxia and the pathophysiology of pulmonary hypertension, detailing the molecular components involved.
  • Line 130: Mechanosensors are mentioned without referring to shear stress on the endothelium, which is fundamental to vasoactive responses. Calcium is also mentioned as a remodeling mechanism, despite the involvement of other factors such as VEGF.
  • Line 145: TRPC is not defined (only TRP is mentioned).
  • Line 155: TRPV is not explained.

It should also be clarified that calcium's physiological response differs depending on whether it is released in the endothelium or smooth muscle cells.

3.2:
This section does not address the pathophysiology.

3.3:
This section focuses on COPD rather than group 3 in general.

  • Line 206: Missing reference.
  • Line 207: When mentioning hypoxia-induced hypertension, specify the type. Different hypoxia models (e.g., hypobaric, normobaric) trigger distinct physiological responses.
  • Line 219: Oxidative stress and its pulmonary effects, particularly in group 3, should be discussed.
  • Line 223: Missing reference.

3.5:
Further elaboration is required for each component. For instance, nitric oxide and prostacyclin are pivotal in treating pulmonary hypertension-associated conditions.

3.6:
Figure 2: The figure's purpose needs clarification. Consider removing or detailing "etc." Additionally, define S100A4 in the figure caption.

3.7:
Mitochondrial dysfunction leading to caspase 3-mediated apoptosis, as seen in certain pathologies, is not discussed.

3.8:
This section discusses treatments in clinical phases. It would be interesting to explore treatments in animal models, such as using melatonin as a free radical scavenger.

3.9:
HIF is a master regulator of pulmonary hypertension. The associated mechanisms should be explored in greater depth.

  • Line 348: Missing reference.

3.11:
Several miRNAs are involved in different PH subtypes. A table summarizing the miRNAs, their targets, and references would be valuable.

4. Figure 3:
Define TR and MR in the figure caption.

5. Table 2:
Indicate the year of the patents.

Figure 4:
This figure is overly basic and does not enhance the review. Clearly identify the therapeutic pathways being targeted with specific objectives.

The review's purpose is to discuss the pathophysiology of group 3, identify the existing challenges, and propose therapeutic targets accordingly.

6. Conclusion:
The conclusion needs improvement. Summarize the developed ideas, explain applications, and propose challenges and future approaches with greater depth.

Author Response

Response to reviewer 2

We greatly appreciate the reviewer’s comments.

  1. Introduction:

Comment: 1. This section discusses pulmonary hypertension; however, it is essential to include the clinical classification of pulmonary hypertension mentioned in section 2.2.

Response: In accordance with the reviewer’s comment, this information is included in section 1.2, Clinical classification of pulmonary hypertension, in the Introduction.

  1. Pulmonary Hypertension:

Comment: The definitions of hypertension and COPD should be included in the introduction section.

Response: In accordance with the reviewer’s comment, this information is included in the Introduction.

  1. Pathogenesis of Group 3 Pulmonary Hypertension:

Comment: Table 1: The table suggests a causal relationship between the pairs listed. For example: inflammation and miRNAs. The table needs improvement.

It is recommended to organize the sections based on causes and consequences. For instance, section 3 could represent causes and section 4 consequences.

Response: Thank you for your comments. Table 1 does not suggest a causal relationship between the pairs listed. The table has therefore been revised. In accordance with the reviewer's comments, the sections have been organized on the basis of cause and effect. Section 3 is the cause and Section 4 is the consequence.

Comment: 3.1: The type of injury related to hypoxia (chronic, acute, or intermittent) should be specified. Additionally, detailed explanations of the translational animal models used, including the duration of hypoxia exposure, are necessary. While vascular remodeling is mentioned as a consequence of HAP, this occurs after prolonged injury periods. Moreover, components of the immune and inflammatory systems via HIF-1α are mentioned without detailing the molecular mechanisms involved.

Response: Thank you for your comments. In accordance with the reviewer's comments, ‘acute’ (line 127)’ and ‘chronic (line 130)’ were added. In addition, several sentences on HIFs were added in section 3.4 (lines 241-257).

Comment: Line 112: Missing reference. Line 119: Missing reference.

Response: Thank you for your comments. In accordance with the reviewer's comments, we have added reference #22 (line 126) and #24-28 (line 134).

Comment: Figure 1: This figure does not contribute to the review. It would be more appropriate to illustrate the relationship between hypoxia and the pathophysiology of pulmonary hypertension, detailing the molecular components involved.

Response: Thank you for your comments. In accordance with the reviewer's comments, we have revised Figure 1.

Comment: Line 130: Mechanosensors are mentioned without referring to shear stress on the endothelium, which is fundamental to vasoactive responses. Calcium is also mentioned as a remodeling mechanism, despite the involvement of other factors such as VEGF.

Response: Thank you for your comments. In accordance with the reviewer's comments, we have added the following sentences: “Elevated pulmonary arterial pressure due to chronic hypoxic pulmonary vasoconstriction increases vessel wall stress and strain and endothelial fluid shear stress. These mechanical cues promote vasoconstriction and vascular remodeling, which exacerbate PH [30].” (lines 138-141) and “The increased expression of TRPC4 in human PAECs exposed to hypoxia was associated with capacitative Ca2+ entry via store-operated Ca2+ channels. This resulted in increased binding to activator protein-1 (AP-1), a transcription factor regulating the expression of genes involved in cell proliferation and migration, such as VEGF and PDGF genes [35].” (lines 172-176).

Comment: Line 145: TRPC is not defined (only TRP is mentioned). Line 155: TRPV is not explained.

Response: In accordance with the reviewer's comments, we have defined TRPC (line 165) and TRPV (line 179).

Comment: It should also be clarified that calcium's physiological response differs depending on whether it is released in the endothelium or smooth muscle cells.

Response: In accordance with the reviewer's comments, we have added several sentences on PAECs and PASMCs (lines 172-176 and 183-188).

Comment: 3.2: This section does not address the pathophysiology.

Response: In accordance with the reviewer's comments, we have added the following sentences: “Cigarette smoke is also associated with mitochondrial fission and fusion imbalances, leading to mitochondrial oxidative stress and functional impairment in rat lung micro-vascular endothelial cells [40]. Cigarette smoke also caused aberrant mitophagy, increased mitochondrial oxidative stress, and reduced mitochondrial respiration. The inhibition of mitochondrial fission and mitochondria-specific antioxidants may be useful therapeutic strategies for cigarette smoke-induced endothelial injury and associated pulmonary diseases [40].” (lines 215-221).

Comment: 3.3: This section focuses on COPD rather than group 3 in general. Line 206: Missing reference. Line 223: Missing reference.

Response: In accordance with the reviewer's comments, we have added reference #16, 17 (line 394) and #82 (line 417).

Comment: Line 207: When mentioning hypoxia-induced hypertension, specify the type. Different hypoxia models (e.g., hypobaric, normobaric) trigger distinct physiological responses.

Response: In accordance with the reviewer's comments, we have added “normobaric” (line395).

Comment: Line 219: Oxidative stress and its pulmonary effects, particularly in group 3, should be discussed.

Response: In accordance with the reviewer's comments, we have added the following sentences: Elevated oxidative stress is a pivotal pathological feature in PH, often caused by chronic hypoxia or sustained low-oxygen conditions [81]. The increase in reactive oxygen species (ROS) is associated with a higher expression of NADPH oxidases in the pulmonary arteries and contributes to vascular remodeling by promoting smooth muscle cell proliferation and fibrosis [81].” (lines 407-411).

Comment: 3.5: Further elaboration is required for each component. For instance, nitric oxide and prostacyclin are pivotal in treating pulmonary hypertension-associated conditions.

Response: In accordance with the reviewer's comments, we have added the following sentences: “A phase III randomized, double-blind, placebo-controlled trial (REBUILD trial) revealed that there was no demonstrable benefit from inhaled nitric oxide in patients with fibrotic interstitial lung disease receiving supplemental oxygen in daily physical activity assessed by means of actigraphy (NCT03267108) [49].” (lines 274-277) and revised Figure 4.

Comment: 3.6: Figure 2: The figure's purpose needs clarification. Consider removing or detailing "etc." Additionally, define S100A4 in the figure caption.

Response: In accordance with the reviewer's comments, we have revised Figure 2 and its caption.

Comment: 3.7: Mitochondrial dysfunction leading to caspase 3-mediated apoptosis, as seen in certain pathologies, is not discussed.

Response: In accordance with the reviewer's comments, we have added the following sentences: “Given that pulmonary arterial remodeling can be caused by the excessive proliferation and apoptosis resistance of PASMCs, the inhibition of cell proliferation or induction of cell apoptosis is considered an efficient therapeutic strategy for PH. For example, Li et al. reported that hypoxia promoted the proliferation of human PASMCs, inhibited the activity of caspase-3, and increased ROS levels, mitochondrial membrane potential, and the expression of HIF-1alpha and PDK4, which induced glycolysis [94]. Sirtuin 6 over-expression could inhibit the proliferation of human PASMCs, as well as increase the apoptosis rate and reduce the levels of ROS, the mitochondrial membrane potential, and the expression of HIF-1alpha and PDK4 in hypoxic human PASMCs.” (lines 463-471).

Comment: 3.8: This section discusses treatments in clinical phases. It would be interesting to explore treatments in animal models, such as using melatonin as a free radical scavenger.

Response: In accordance with the reviewer's comments, we have added the following sentences: “Antioxidant treatments have been tested in experimental animal models. For ex-ample, Huang et al. reported that melatonin treatment significantly attenuated the levels of right ventricular systolic pressure and oxidative and inflammatory markers in hypoxic animals with a marked increase in eNOS phosphorylation in the lungs [58].” (line 327-330).

Comment: 3.9: HIF is a master regulator of pulmonary hypertension. The associated mechanisms should be explored in greater depth.

Response: Thank you for your comments. In accordance with the reviewer's comments, we have added several sentences in section 3.4 HIFs (lines 241-257).

Comment: Line 348: Missing reference.

Response: In accordance with the reviewer's comments, we have added reference #53 (line 310).

Comment: 3.11: Several miRNAs are involved in different PH subtypes. A table summarizing the miRNAs, their targets, and references would be valuable.

Response: In accordance with the reviewer's comments, we have added Table 2.

Comment: 4. Figure 3: Define TR and MR in the figure caption.

Response: In accordance with the reviewer's comments, we have defined TR and MR.

Comment: 5. Table 2: Indicate the year of the patents.

Response: Thank you for your comments. We are sorry, no patents have yet been identified in ongoing research.

Comment: Figure 4: This figure is overly basic and does not enhance the review. Clearly identify the therapeutic pathways being targeted with specific objectives. The review's purpose is to discuss the pathophysiology of group 3, identify the existing challenges, and propose therapeutic targets accordingly.

Response: In accordance with the reviewer's comments, we have revised Figure 4.

  1. Conclusion:

Comment: The conclusion needs improvement. Summarize the developed ideas, explain applications, and propose challenges and future approaches with greater depth.

Response: In accordance with the reviewer's comments, we have added the following sentences: “Several factors have been reported to mediate vascular remodeling induced by hypoxic pulmonary vasoconstriction, such as HIF-1α and mechanosensors, including TRP channels. Current efforts are focused on new target molecules, such as mechanoreceptors that sense pulmonary vasoconstriction. These could lead to new therapies to inhibit vascular remodeling in the future.” (lines598-602).

Reviewer 3 Report (New Reviewer)

Comments and Suggestions for Authors

1-The review mentions the role of microRNAs and genetic predisposition, it does not extensively cover the latest research on potential biomarkers for early diagnosis or prognosis of Group 3 pulmonary hypertension. Incorporate recent findings on genetic markers, epigenetic changes, and circulating biomarkers that can help stratify patients better and predict

2-The interplay of pulmonary hypertension with other conditions such as heart failure, diabetes, or metabolic syndrome is not sufficiently addressed, even though these are common in affected populations. A dedicated section could explore how co-existent diseases contribute to the severity and management of Group 3 pulmonary hypertension

3-While several key pathways have been outlined, the interactions and crosstalk between various mechanisms such as oxidative stress, inflammation, and hypoxia-inducible factors are not sufficiently detailed. Please can you provide a more integrative view of how these pathways interact in contributing to vascular remodeling and pulmonary hypertension.

Author Response

Response to reviewer 3

We greatly appreciate the reviewer’s comments.

Comment: 1-The review mentions the role of microRNAs and genetic predisposition, it does not extensively cover the latest research on potential biomarkers for early diagnosis or prognosis of Group 3 pulmonary hypertension. Incorporate recent findings on genetic markers, epigenetic changes, and circulating biomarkers that can help stratify patients better and predict.

Response: Thank you for your comment. In accordance with the reviewer’s comment, we have added the following sentences: “Jin et al. reported that blood hsa_circNFXL1_009 levels were reduced in COPD patients with PH and that the administration of exogenous hsa_circNFXL1_009 attenuated the hypoxia-induced proliferation, apoptotic resistance, and migration of human PASMCs [102].” (lines 493-496), “Zhou et al. reported that hsa_circ_0016070 expression was significantly increased in the lungs of COPD patients with PH and that hsa_circ_0016070 promoted PASMC proliferation by regulating miR-942/CCND1 [103]” ‘lines 497-499), “ErbB3 expression levels in serum were significantly correlated with mPAP, PVR, or the cardiac index.” (lines 286-287) and “ErbB3 serves as a novel biomarker and a promising target for the treatment of hypoxia-induced PH.” (lines 295-296).

Comment: 2-The interplay of pulmonary hypertension with other conditions such as heart failure, diabetes, or metabolic syndrome is not sufficiently addressed, even though these are common in affected populations. A dedicated section could explore how co-existent diseases contribute to the severity and management of Group 3 pulmonary hypertension.

Response: Thank you for your comment. In accordance with the reviewer’s comment, we have added new section “3.9. Comorbidities” and the following sentences: “Comorbidities impact a large proportion of patients with COPD and ILD. Comorbidities include allergic disease, coronary heart disease, congestive heart failure, diabetes, metabolic syndrome, and sleep apnea, which affect pulmonary hypertension and its prognosis [75, 76].” (lines 368-372).

Comment: 3-While several key pathways have been outlined, the interactions and crosstalk between various mechanisms such as oxidative stress, inflammation, and hypoxia-inducible factors are not sufficiently detailed. Please can you provide a more integrative view of how these pathways interact in contributing to vascular remodeling and pulmonary hypertension.

Response: Thank you for your comment. In accordance with the reviewer’s comment, we have revised Figure 1 and have restructured the sections and paragraphs according to the order in Table 1.

Round 2

Reviewer 1 Report (New Reviewer)

Comments and Suggestions for Authors

Thank you for addressing all of my concerns despite the low novelty of the study, this manuscript could be acceptable in this current version

Author Response

Comment: Thank you for addressing all of my concerns despite the low novelty of the study, this manuscript could be acceptable in this current version.

Response: We greatly appreciate the reviewer’s comments.

Reviewer 2 Report (New Reviewer)

Comments and Suggestions for Authors

Line 36: "Lung disease, particularly chronic obstructive pulmonary disease (COPD), is the second most common cause" does not indicate prevalence. Include a discussion of the pathophysiology, as the review covers COPD.

Line 80: Remove the objectives title. It is unnecessary to include a separate section for this.

Section 3.1: Provide a detailed explanation of the relationship between shear stress and the function of mechanosensors, which modifies vascular homeostasis. (PMID: 12577139; PMID: 21733876)

Lines 167 and 171: Correct "Ca+2 to Ca+2" (unclear wording or formatting issue).

Lines 207-209: No reference is provided.

Line 248: The information in Section 3.5 and subsections 3.5.1, 3.5.2, and 3.5.3 is insufficient. Previous sections explain the mechanisms in greater detail, whereas here only a therapy is mentioned. There is extensive information available on these molecules, and further elaboration is necessary.
Example: Antioxidant therapies can induce the expression of prostanoids (PMID: 33766627).

Line 311: Additional examples can be included (e.g., PMID: 31583753).

Line 435: Specify the type of apoptosis induced by mitochondrial dysfunction, such as the intrinsic pathway.

Line 562: Include the year of each patent in the table.

Author Response

We greatly appreciate the reviewer’s comments.

Comment: Line 36: "Lung disease, particularly chronic obstructive pulmonary disease (COPD), is the second most common cause" does not indicate prevalence. Include a discussion of the pathophysiology, as the review covers COPD.

Response: In accordance with the reviewer’s comment, we have deleted “particularly chronic obstructive pulmonary disease (COPD)”.

Comment: Line 80: Remove the objectives title. It is unnecessary to include a separate section for this.

Response: In accordance with the reviewer’s comment, we have deleted the objectives title “1.3. Aims of this review”.

Comment: Section 3.1: Provide a detailed explanation of the relationship between shear stress and the function of mechanosensors, which modifies vascular homeostasis. (PMID: 12577139; PMID: 21733876).

Response: In accordance with the reviewer's comments, we have added the following sentence: “Low shear stress, or changes in shear stress direction as seen in turbulence, promote endothelial proliferation and apoptosis, shape changes, vasoconstriction, coagulation, and secretion of substances that promote platelet aggregation [31, 32].” (page 4, line 6).

Comment: Lines 167 and 171: Correct "Ca+2 to Ca+2" (unclear wording or formatting issue).

Response: Thank you for your comments. We have corrected the word to “Ca2+” (page 5).

Comment: Lines 207-209: No reference is provided.

Response: In accordance with the reviewer's comments, we have added reference #43 (page 6, line 8).

Comment: Line 248: The information in Section 3.5 and subsections 3.5.1, 3.5.2, and 3.5.3 is insufficient. Previous sections explain the mechanisms in greater detail, whereas here only a therapy is mentioned. There is extensive information available on these molecules, and further elaboration is necessary.

Example: Antioxidant therapies can induce the expression of prostanoids (PMID: 33766627).

Response: In accordance with the reviewer's comments, we have added the following sentences: “However, drugs targeting these three pathways are not approved for patients with COPD-associated PH due to lack of evidence [50].” (page 7, line 1 in Section 3.5), “Stolz et al. reported that the oral administration of the endothelin receptor antagonist bosentan not only failed to improve exercise capacity but also deteriorated hypoxaemia and functional status in severe chronic obstructive pulmonary disease patients without severe pulmonary hypertension at rest [53]. In contrast, Valerio et al. reported bosentan treatment resulted in a significant improvement of PAP, PVR and 6-min walk distance (6MWD) [54]. Thus, there are still insufficient data to fully support the effects of endothelin receptor antagonists (ERAs) on pulmonary hemodynamics and exercise tolerance in patients with COPD-PH.” (page 7, line 7 in subsection 3.5.1), “Hypoxia has been shown to reduce eNOS expression and nitric oxide production in the lungs of newborn piglets exposed to chronic hypoxia [55].” (page 7, line 17 in subsection 3.5.2), and “Pulmonary arterial hypertension of the newborn is a syndrome caused by chronic hypoxia, characterized by decreased vasodilator function, a marked vasoconstrictor activity, and proliferation of smooth muscle cells in the pulmonary circulation. Antioxidant therapy with melatonin induces prostanoid expression in neonates [58].” (page 7, line 26 in subsection 3.5.3).

Comment: Line 311: Additional examples can be included (e.g., PMID: 31583753).

Response: In accordance with the reviewer's comments, we have added the following sentence: “Gonzalez-Candia et al reported that melatonin had long-lasting beneficial effects on pulmonary vascular reactivity and redox balance in chronic hypoxia in newborn lambs gestated and born at 3600 m [67].” (page 8, line 32 in section 3.7).

Comment: Line 435: Specify the type of apoptosis induced by mitochondrial dysfunction, such as the intrinsic pathway.

Response: In accordance with the reviewer's comments, we have added the regarding apoptosis and mitochondrial dysfunction: “Bhansali et al reported an excessive rise in mtROS production and disrupted membrane potential, accompanied by enhanced DNA damage and reduced autophagy was observed, highlighting the 'apoptosis resistance’ phenotype, in hypoxic PASMCs [102].” (page 11, line 9 in section 4.3).

Comment: Line 562: Include the year of each patent in the table.

Response: Thank you for your comments. We are sorry, but these studies are ongoing, and no patents have been identified yet.

This manuscript is a resubmission of an earlier submission. The following is a list of the peer review reports and author responses from that submission.

Round 1

Reviewer 1 Report

Comments and Suggestions for Authors

The Authors did not answer to my previous concerns. The very minor changes to the text do not meet any of the open questions. Likewise, the single small change to Figure 1 does not meet the need to make a serious attempt to synthesize the observations and to propose some hypotheses to address future research. The Authors’ reply to the other comments answer questions different from those that formed my previous concerns. Therefore, all my previous concerns remain. 

Comments on the Quality of English Language

English is almost fine

Author Response

Comment: The Authors did not answer to my previous concerns. The very minor changes to the text do not meet any of the open questions. Likewise, the single small change to Figure 1 does not meet the need to make a serious attempt to synthesize the observations and to propose some hypotheses to address future research. The Authors’ reply to the other comments answer questions different from those that formed my previous concerns. Therefore, all my previous concerns remain. 

Response: Thank you for your comment. We agree with the reviewer’s opinion. We added the following sentences: “5. Clinical implications

Data supporting the use of PAH therapy in group 3 PH are limited and require further study. Vasoactive medications may be effective in patients with severe hemodynamic parenchymal lung disease, and these patients may be a target population for future studies.

The INCREASE trial (NCT02630316) evaluated the safety and efficacy of inhaled Treprostinil, an analogue of prostacyclin, in patients with PH due to ILD [99]. In this patient population, inhaled treprostinil improved exercise capacity from baseline, assessed with the use of a 6-minute walk test, as compared with placebo. In addition, treatment with inhaled treprostinil was associated with a lower risk of clinical worsening than that with placebo, a reduction in NT-proBNP levels, and fewer exacerbations of underlying lung disease. The PERFECT study (NCT03496623) evaluated the safety and efficacy of inhaled treprostinil in PH associated with COPD [100]. Results showed that numerically more adverse events, deaths, treatment discontinuations, and study discontinuations occurred in patients receiving inhaled treprostinil compared to placebo. In addition, patients receiving inhaled treprostinil did not show an improvement in 6MWD compared to placebo. Overall, the trial showed that the risks in treating PH-COPD patients with inhaled treprostinil outweighed the potential positive benefits, thereby justifying early discontinuation. Thus, the efficacy and safety of the same drug varied depending on the type of parenchymal lung disease. These studies suggest that future clinical trials for group 3 PH will require disease-specific and fine-tuned designs. It is hoped that future clinical studies targeting the small molecules described in this review will draw on the lessons learned from these studies.

In the treatment of group 1 PAH, a new treatment with sotatercept, a fusion protein that traps activins and growth differentiation factors, has begun [101]. Treatment of group 3 PH with selective pulmonary vasodilators such as PGI2 analog is currently being attempted. In the future, it is expected that therapies targeting novel molecules such as those described in this review will be developed (Figure 4)” (page 11, lines 464-491) and Figure 4.

Reviewer 2 Report

Comments and Suggestions for Authors

Your paper does a great job diving into the complex world of Group 3 pulmonary hypertension. The mechanisms are well-explained, and it's clear you've put a lot of thought into detailing the various factors at play. Here are just a few things that might make it even stronger:

  1. Connect the dots to clinical impact: You’ve explained the science beautifully, but it could help to spell out more clearly how these insights might translate into real-world treatments. Clinicians will appreciate knowing what this means for patient care.

  2. Simplify where you can: Some sections, particularly those on molecular pathways, are quite technical. Maybe a diagram or breaking down some of the language would help make it easier for readers to follow.

  3. Minor language tweaks: The English is solid overall, but there are a few areas where simplifying sentence structure could improve the flow and readability.

Comments on the Quality of English Language

The English in your paper is quite strong overall. It reads well, and the key ideas come through clearly. That said, there are a few spots where the sentences could flow more smoothly. Some minor edits—like simplifying complex phrases or breaking up longer sentences—could make it even easier for readers to follow along. But overall, it’s well-written and just needs a little polishing to make it perfect!

Author Response

We greatly appreciate the reviewer’s comments.

Comment: #1. Connect the dots to clinical impact: You’ve explained the science beautifully, but it could help to spell out more clearly how these insights might translate into real-world treatments. Clinicians will appreciate knowing what this means for patient care.

Response: Thank you for your comment. In accordance with the reviewer’s comment, we added the following sentences: “5. Clinical implications

Data supporting the use of PAH therapy in group 3 PH are limited and require further study. Vasoactive medications may be effective in patients with severe hemodynamic parenchymal lung disease, and these patients may be a target population for future studies.

The INCREASE trial (NCT02630316) evaluated the safety and efficacy of inhaled Treprostinil, an analogue of prostacyclin, in patients with PH due to ILD [99]. In this patient population, inhaled treprostinil improved exercise capacity from baseline, assessed with the use of a 6-minute walk test, as compared with placebo. In addition, treatment with inhaled treprostinil was associated with a lower risk of clinical worsening than that with placebo, a reduction in NT-proBNP levels, and fewer exacerbations of underlying lung disease. The PERFECT study (NCT03496623) evaluated the safety and efficacy of inhaled treprostinil in PH associated with COPD [100]. Results showed that numerically more adverse events, deaths, treatment discontinuations, and study discontinuations occurred in patients receiving inhaled treprostinil compared to placebo. In addition, patients receiving inhaled treprostinil did not show an improvement in 6MWD compared to placebo. Overall, the trial showed that the risks in treating PH-COPD patients with inhaled treprostinil outweighed the potential positive benefits, thereby justifying early discontinuation. Thus, the efficacy and safety of the same drug varied depending on the type of parenchymal lung disease. These studies suggest that future clinical trials for group 3 PH will require disease-specific and fine-tuned designs. It is hoped that future clinical studies targeting the small molecules described in this review will draw on the lessons learned from these studies.

In the treatment of group 1 PAH, a new treatment with sotatercept, a fusion protein that traps activins and growth differentiation factors, has begun [101]. Treatment of group 3 PH with selective pulmonary vasodilators such as PGI2 analog is currently being attempted. In the future, it is expected that therapies targeting novel molecules such as those described in this review will be developed (Figure 4)” (page 11, lines 464-491) and Figure 4.

Comment: #2. Simplify where you can: Some sections, particularly those on molecular pathways, are quite technical. Maybe a diagram or breaking down some of the language would help make it easier for readers to follow.

Response: Thank you for your comment. In accordance with the reviewer’s comment, we added Figure 2 and Figure 4.

Comment: #3. Minor language tweaks: The English is solid overall, but there are a few areas where simplifying sentence structure could improve the flow and readability.

Response: Thank you for your comment. In accordance with the reviewer’s comment, We hired SES Translation and Proofreading Service to proofread our English text.

Reviewer 3 Report

Comments and Suggestions for Authors

The manuscript, "Pathophysiology of Group 3 Pulmonary Hypertension Associated with Lung Diseases and/or Hypoxia," provides a detailed and comprehensive review of the molecular mechanisms underlying Group 3 pulmonary hypertension (PH). The authors have effectively summarized key aspects such as hypoxic pulmonary vasoconstriction, pulmonary vascular remodelling, and the roles of various molecular pathways like TRPC channels, oxidative stress, and mitochondrial dysfunction.

However, several areas need improvement to increase the manuscript's scientific impact and relevance.

First, the manuscript lacks novelty, as it mainly reviews well-established mechanisms without presenting new insights or hypotheses.

Additionally, there is an overemphasis on molecular details without sufficient integration of clinical relevance, limiting its appeal to a broader audience that includes clinicians. The discussion of therapeutic implications is also underdeveloped, particularly regarding emerging treatments or potential interventions targeting the described pathways.

Furthermore, the manuscript would benefit from a more structured comparative analysis of the different mechanisms, highlighting their relative importance and potential as therapeutic targets.

Consider adding more recent studies to the reference list to reflect the latest advancements in the field.

The figures and illustrations are clear, but they are highly focused on basic science. There are no visual aids that convey clinical implications or patient outcomes, which could bridge the gap between basic science and clinical relevance. Including figures that connect molecular mechanisms to potential clinical pathways would improve the manuscript's applicability to broader audiences.

Author Response

We greatly appreciate the reviewer’s comments.

Comment: #1. First, the manuscript lacks novelty, as it mainly reviews well-established mechanisms without presenting new insights or hypotheses. 

Response: Thank you for your comment. We agree with the reviewer’s opinion. In accordance with the reviewer’s comment, we add the following sentences: “In the treatment of group 1 PAH, a new treatment with sotatercept, a fusion protein that traps activins and growth differentiation factors, has begun [101]. Treatment of group 3 PH with selective pulmonary vasodilators such as PGI2 analog is currently being attempted. In the future, it is expected that therapies targeting novel molecules such as those described in this review will be developed (Figure 4)” (page 11, lines 487-491) and Figure 4.

Comment: #2. Additionally, there is an overemphasis on molecular details without sufficient integration of clinical relevance, limiting its appeal to a broader audience that includes clinicians. The discussion of therapeutic implications is also underdeveloped, particularly regarding emerging treatments or potential interventions targeting the described pathways.

Response: Thank you for your comment. In accordance with the reviewer’s comment, we add the following sentences: “5. Clinical implications

Data supporting the use of PAH therapy in group 3 PH are limited and require further study. Vasoactive medications may be effective in patients with severe hemodynamic parenchymal lung disease, and these patients may be a target population for future studies.

The INCREASE trial (NCT02630316) evaluated the safety and efficacy of inhaled Treprostinil, an analogue of prostacyclin, in patients with PH due to ILD [99]. In this patient population, inhaled treprostinil improved exercise capacity from baseline, assessed with the use of a 6-minute walk test, as compared with placebo. In addition, treatment with inhaled treprostinil was associated with a lower risk of clinical worsening than that with placebo, a reduction in NT-proBNP levels, and fewer exacerbations of underlying lung disease. The PERFECT study (NCT03496623) evaluated the safety and efficacy of inhaled treprostinil in PH associated with COPD [100]. Results showed that numerically more adverse events, deaths, treatment discontinuations, and study discontinuations occurred in patients receiving inhaled treprostinil compared to placebo. In addition, patients receiving inhaled treprostinil did not show an improvement in 6MWD compared to placebo. Overall, the trial showed that the risks in treating PH-COPD patients with inhaled treprostinil outweighed the potential positive benefits, thereby justifying early discontinuation. Thus, the efficacy and safety of the same drug varied depending on the type of parenchymal lung disease. These studies suggest that future clinical trials for group 3 PH will require disease-specific and fine-tuned designs. It is hoped that future clinical studies targeting the small molecules described in this review will draw on the lessons learned from these studies.” (page 11, lines 464-486)

Comment: #3. Furthermore, the manuscript would benefit from a more structured comparative analysis of the different mechanisms, highlighting their relative importance and potential as therapeutic targets. 

Response: Thank you for your comment. In accordance with the reviewer’s comment, we add Figure 4.

Comment: #4. Consider adding more recent studies to the reference list to reflect the latest advancements in the field.

Response: Thank you for your comment. In accordance with the reviewer’s comment, we add references 99, 100 and 101.

Comment: #5. The figures and illustrations are clear, but they are highly focused on basic science. There are no visual aids that convey clinical implications or patient outcomes, which could bridge the gap between basic science and clinical relevance. Including figures that connect molecular mechanisms to potential clinical pathways would improve the manuscript's applicability to broader audiences.

Response: Thank you for your comment. In accordance with the reviewer’s comment, we add Figure 4.

Round 2

Reviewer 2 Report

Comments and Suggestions for Authors
  • Enhancing the discussion on how these mechanisms could inform potential treatments or ongoing trials for Group 3 PH would make the review more impactful.
  • Consider tightening sections on oxidative stress and TRP channels to improve readability and thematic continuity.
  • A brief outline of research gaps and potential directions would give readers a clearer roadmap for advancing knowledge in Group 3 PH.
Comments on the Quality of English Language
  1. Overall, the language is clear and well-structured, though a few sections, especially in complex areas like oxidative stress and hypoxia, could benefit from more straightforward language to enhance readability.

  2. Some terminology, such as specific cellular mechanisms (e.g., "endothelial-to-mesenchymal transition," "hypoxic pulmonary vasoconstriction") is well-defined but might benefit from occasional simplification or brief rephrasing to improve comprehension for a broader audience.

  3. There are minimal grammatical errors; however, a few sentences would benefit from slight rephrasing to improve flow. For instance, instead of "In pulmonary hypertension, the main factor causing right heart failure is increased afterload," consider "The primary factor leading to right heart failure in pulmonary hypertension is increased afterload."

  4. Some sections contain redundant phrases that could be streamlined for brevity, which would enhance overall readability and keep readers engaged.

  5. In a few areas, the passive voice slightly impedes clarity. For example, rephrasing "was reported to be poor prognostic factors" to "serve as poor prognostic factors" would make the text more direct and engaging.

Author Response

We greatly appreciate the reviewer’s comments.

Comments and Suggestions for Authors

Comment: #1. Enhancing the discussion on how these mechanisms could inform potential treatments or ongoing trials for Group 3 PH would make the review more impactful.

Response: Thank you for your comment. In accordance with the reviewer’s comment, we add the following sentences:“Table 2 shows ongoing trials evaluating the effects of various PH therapies in patients with PH associated with COPD. Changes in mitochondrial metabolic parameters as assessed by the Agilent Seahorse extracellular flux bioanalyzer will be evaluated after pioglitazone administration (NCT0633679). We look forward to future development of therapies targeting other molecular biological mechanisms discussed in this review.” (page 11, lines 461-465) and “Table 2. Ongoing Trials Evaluating the Effect of Various Therapies in Patients with PH associated with COPD” (pages 11-12).

Comment: #2. Consider tightening sections on oxidative stress and TRP channels to improve readability and thematic continuity.

Response: Thank you for your comment. In accordance with the reviewer’s comment, we deleted several sentences and tightened the sections on oxidative stress and TRP channels (pages 4 and 8).

Comment: #3. A brief outline of research gaps and potential directions would give readers a clearer roadmap for advancing knowledge in Group 3 PH.

Response: Thank you for your comment. In accordance with the reviewer’s comment, we add the following sentences: “In the future, in addition to selective pulmonary vasodilators, we hope to develop treatments for group 3 PH that target novel molecules that improve vascular remodeling (Figure 4).” (page 12, lines 472-474).

Comments on the Quality of English Language

Comment: #1. Overall, the language is clear and well-structured, though a few sections, especially in complex areas like oxidative stress and hypoxia, could benefit from more straightforward language to enhance readability.

Response: Thank you for your comment. To improve readability, we deleted some sentences and used abbreviations(pages 4 and 8).

Comment: #2. Some terminology, such as specific cellular mechanisms (e.g., "endothelial-to-mesenchymal transition," "hypoxic pulmonary vasoconstriction") is well-defined but might benefit from occasional simplification or brief rephrasing to improve comprehension for a broader audience.

Response: Thank you for your comment. In accordance with the reviewer’s comment, we added several abbreviations (ROS, SMC, PAEC and PASMC).

Comment: #3. There are minimal grammatical errors; however, a few sentences would benefit from slight rephrasing to improve flow. For instance, instead of "In pulmonary hypertension, the main factor causing right heart failure is increased afterload," consider "The primary factor leading to right heart failure in pulmonary hypertension is increased afterload."

Response: Thank you for your comment. In accordance with the reviewer’s comment, we rephrased the sentence (page 10, line 429).

Comment: #4. Some sections contain redundant phrases that could be streamlined for brevity, which would enhance overall readability and keep readers engaged.

Response: Thank you for your comment. In accordance with the reviewer’s comment, we avoided redundancy by using some abbreviations.

Comment: #5. In a few areas, the passive voice slightly impedes clarity. For example, rephrasing "was reported to be poor prognostic factors" to "serve as poor prognostic factors" would make the text more direct and engaging.

Response: Thank you for your comment. In accordance with the reviewer’s comment, we rephrased the sentence (page 2, line 70) and removed several passive phrases (pages 4 and 8).

Reviewer 3 Report

Comments and Suggestions for Authors

In this revised manuscript, the authors have improved the quality of their review by making all the advised changes and addressing the comments.

Round 3

Reviewer 2 Report

Comments and Suggestions for Authors

This is an impressive and comprehensive review on pulmonary hypertension associated with lung disease and hypoxia. Your work covers the key mechanisms effectively and is a valuable contribution to the field.

The abstract is clear but could be more focused by emphasizing the main findings to engage readers right from the start. The introduction provides a solid background, though condensing some parts would help keep the reader’s attention on the main objective.

Your detailed sections on hypoxic pulmonary vasoconstriction and vascular remodeling are excellent. Adding a few sentences on the potential clinical impact of these mechanisms would connect the science to practice, making it even more impactful.

The figures and tables are well-chosen and support your narrative. Double-check that they are labeled clearly and formatted uniformly. The conclusion is concise, but a brief mention of future research directions or clinical applications could add depth.

Lastly, make sure all references are formatted properly and follow journal guidelines, particularly listing authors before "et al."

Overall, this is a well-written manuscript that I believe, with minor refinements, will greatly benefit readers interested in the evolving understanding of pulmonary hypertension.

Comments on the Quality of English Language

na